# Response Prediction and Evaluation Using PET in Patients with Solid Tumors Treated with Immunotherapy

**DOI:** 10.3390/cancers13123083

**Published:** 2021-06-21

**Authors:** Frank J. Borm, Jasper Smit, Daniela E. Oprea-Lager, Maurits Wondergem, John B. A. G. Haanen, Egbert F. Smit, Adrianus J. de Langen

**Affiliations:** 1Department of Pulmonary Diseases, Leiden University Medical Centre, 2333 ZA Leiden, The Netherlands; f.borm@nki.nl (F.J.B.); ja.smit@nki.nl (J.S.); 2Department of Thoracic Oncology, NKI-AvL, 1066 CX Amsterdam, The Netherlands; e.smit@nki.nl; 3Department of Radiology & Nuclear Medicine, Cancer Center Amsterdam, Amsterdam University Medical Centers (Location VU University Medical Center), 1081 HV Amsterdam, The Netherlands; d.oprea-lager@amsterdamumc.nl; 4Department of Nuclear Medicine, NKI-AvL, 1066 CX Amsterdam, The Netherlands; m.wondergem@nki.nl; 5Department of Medical Oncology, NKI-AvL, 1066 CX Amsterdam, The Netherlands; j.haanen@nki.nl

**Keywords:** immunotherapy, positron emission tomography, NSCLC, melanoma, biomarker

## Abstract

**Simple Summary:**

In cancer treatment, immunotherapy is increasingly becoming important as a component of first-line treatment and has improved the prognosis of patients since its introduction. A large group of patients, however, do not respond to immunotherapy, and predicting a treatment response remains challenging. Furthermore, evaluating a response using conventional computed tomography (CT) scans is not straightforward due to the different mechanism of action of immunotherapy compared to chemotherapy. This review provides an overview of positron emission tomography (PET) in predicting and evaluating treatment response to immunotherapy.

**Abstract:**

In multiple malignancies, checkpoint inhibitor therapy has an established role in the first-line treatment setting. However, only a subset of patients benefit from checkpoint inhibition, and as a result, the field of biomarker research is active. Molecular imaging with the use of positron emission tomography (PET) is one of the biomarkers that is being studied. PET tracers such as conventional ^18^F-FDG but also PD-(L)1 directed tracers are being evaluated for their predictive power. Furthermore, the use of artificial intelligence is under evaluation for the purpose of response prediction. Response evaluation during checkpoint inhibitor therapy can be challenging due to the different response patterns that can be observed compared to traditional chemotherapy. The additional information provided by PET can potentially be of value to evaluate a response early after the start of treatment and provide the clinician with important information about the efficacy of immunotherapy. Furthermore, the use of PET to stratify between patients with a complete response and those with a residual disease can potentially guide clinicians to identify patients for which immunotherapy can be discontinued and patients for whom the treatment needs to be escalated. This review provides an overview of the use of positron emission tomography (PET) to predict and evaluate treatment response to immunotherapy.

## 1. Introduction

The treatment of many advanced-stage solid malignancies, especially non-small cell lung cancer (NSCLC), renal cell carcinoma, and melanoma, has changed dramatically since the introduction of immunotherapy. Anti-programmed cell death protein 1/programmed cell death protein ligand (PD1/PD-L1) antibodies, whether or not in combination with a cytotoxic T-lymphocyte antigen 4 (CTLA-4) antibody, chemotherapy, and/or anti-angiogenic therapy, currently have an established role in the first-line treatment setting and improve overall survival [1,2,3,4,5].

Despite this progress, only a subset of patients benefits from immune checkpoint inhibition, while a substantial proportion of patients do not [1,4,5,6,7,8,9,10,11,12]. Since the clinical status of patients can deteriorate quickly during a non-effective treatment, choosing the most effective upfront treatment is of utmost importance. Consequently, the field of biomarker research is very active, but predicting a response remains challenging. PD-L1 immunohistochemistry (IHC) is currently the most widely used biomarker to select patients for checkpoint inhibitor therapy. However, it is far from perfect as responses are observed in patients without tumor PD-L1 expression, while patients with PD-L1 expressing tumors often fail to respond. The low predictive value could be caused by PD-L1 expression heterogeneity that cannot be fully captured by a small biopsy [13,14,15]. Other biomarkers are actively being explored. One example of an alternative biomarker is tumor mutational burden (TMB), derived from either blood or tissue. This biomarker is able to predict response, irrespective of PD-L1 status, but failed to outperform PD-L1 IHC [16,17].

In contrast to tissue-based biomarkers, whole-body molecular imaging is a non-invasive technique that enables whole-body target quantification, including the primary tumor and metastases. With the use of positron emission tomography (PET), various tracers can be deployed to visually detect and semi-quantitatively quantify target expression and monitor drug distribution [18,19]. Serial imaging during therapy makes it possible to monitor dynamic changes induced by anticancer therapy that can be used for response evaluation.

Currently, the response is being evaluated with computed tomography (CT) using the Response Evaluation Criteria in Solid Tumors (RECIST) 1.1 [20]. However, RECIST 1.1 has shown its shortcomings. A phenomenon called pseudoprogression (an initial increase in tumor size, followed by a subsequent decrease) can be seen in up to 10% of patients with melanoma and 0–5% of NSCLC patients treated with immune checkpoint inhibitor therapy [21,22,23,24,25]. According to RECIST 1.1, these patients can be misinterpreted as having progressive disease. As a result of this limitation, the immune RECIST (iRECIST) has been developed [26], aiming to correct for pseudoprogression. With this criteria, tumor growth or the appearance of new lesions needs to be confirmed by a subsequent CT scan when patients remain in suitable clinical condition. As pseudoprogression is uncommon, most patients with such a response pattern will have progressive disease, and these patients will have a delay in switching systemic therapy and may not receive an additional line of therapy due to clinical deterioration. Due to the limitations of CT-based response evaluation during immunotherapy, there is a unique opportunity for molecular imaging to improve response evaluation and prediction by providing additional molecular data. In this critical review, we will address the value of PET in patients with solid tumors treated with immune checkpoint inhibitor therapy. We will focus on prediction and evaluation of response with ^18^F-FDG-PET, immune-PET, and the possible predictive value of radiomics.

An extensive search on Pubmed for clinical trials concerning PET and checkpoint inhibitors in solid tumors was performed between January 2015 and December 2020. The most relevant clinical research topics were then extracted, based on the expertise of the authors, with emphasis on the value of PET in patients with solid tumors treated with immune checkpoint inhibitor therapy. Furthermore, prediction and evaluation of response with 18F-FDG-PET, prediction of response with immune-PET, and the possible predictive power of radiomics in PET were assessed. We selected English-written clinical articles where the study was designed to evaluate or predict response and/or immune tracers were evaluated in humans.

## 2. Prediction of Response with PET Using Various Tracers

### 2.1. ^18^F-FDG-PET-CT

Fluor-18-deoxyglucose positron emission tomography (^18^F-FDG-PET) is a widely available and routinely performed procedure. ^18^F-FDG-PET is able to quantify (glucose) metabolism and, because of the combination of PET and CT, it combines information of metabolism with morphological data. Since PD-L1 IHC is currently the only used biomarker in the clinic, the correlation between ^18^F-FDG uptake in tumors and PD-L1 IHC has been extensively studied. Maximum standardized uptake value (SUVmax) has been found to be higher in PD-L1 positive tumors as compared to PD-L1 negative tumors [27,28]. Furthermore, a correlation has been found between ^18^F-FDG uptake and the expression of immune markers such as PD-1, CD8, and CD68 [29,30]. In addition, the accumulation of ^18^F-FDG is relatively higher in macrophages and young granulation tissue compared to most tumor cells [31]. These features indicate that ^18^F-FDG-PET can possibly be used to assess the immunological tumor microenvironment and therefore predict response to immunotherapy.

Multiple studies have looked into the value of different ^18^F-FDG-PET derived parameters in predicting checkpoint inhibitor treatment response [32,33,34,35,36,37,38]. Grizzi et al. reported the preliminary analysis of 27 patients with NSCLC treated with checkpoint inhibitors [32]. They observed that almost all ‘fast progressors’ (8 out of 9) after 8 weeks of treatment had tumor lesions with a SUVmax ≤ 17.1 and a SUVmean ≤ 8.3 at baseline. These cutoff values of SUVmax and SUVmean had a high sensitivity to predict for progressive disease (88.9% and 100%), but the specificity was disappointing (38.9% and 33.3%). These results are comparable with the observations of Takada et al. In 89 patients with NSCLC (treated with nivolumab or pembrolizumab), a cutoff value of SUVmax was identified as 11.1 for patient stratification. The response rate of patients with a SUVmax above 11.1 was significantly higher compared to patients with a SUVmax below 11.1 (RR 41.3% vs. 11.6%, *p* = 0.0012) [38]. These findings suggest that a higher tumor ^18^F-FDG uptake expressed as SUVmax predicts response to checkpoint inhibitor therapy, which can be explained by the positive correlation between PD-L1 expression and ^18^F-FDG uptake [27,28].

An approach to assess both size and uptake of malignant lesions on ^18^F-FDG-PET is to determine total lesion glycolysis (TLG) or metabolic tumor volume (MTV). MTV refers to the metabolically active volume of a tumor lesion, and TLG is the product of SUVmean and MTV and uses both volumetric and metabolic information of the ^18^F-FDG-PET.

Seban et al. [34] studied 80 patients with NSCLC treated with nivolumab. They concluded that a total MTV of all tumor lesions >75 cm^3^ is strongly associated with shorter OS (HR 2.5, 95%CI 1.3–4.7 and HR 3.3, 95%CI 1.6–6.4). Furthermore, Ito et al. [36] showed that high TLG, mainly driven by high MTV, correlates with a worse outcome in patients treated with nivolumab, and this result was also confirmed by the study of Hashimoto et al. [37]. This corresponds with the observations of Evangelista et al. [35]. They assessed the ^18^F-FDG uptake of the entire tumor burden in 32 patients with NSCLC and observed a higher total tumor lesion SUVmax (sum of the individual SUVmax of all measurable lesions) in patients without a response to nivolumab therapy compared to patients with a treatment response.

These results show us that using ^18^F-FDG-PET-CT to predict response to immunotherapy is not straightforward. An important shortcoming of this imaging technique is that glucose metabolism is not specific for either tumor cells or immune cells, let alone differentiate between CD8+ effector T cells and immunosuppressive regulatory T cells. In addition, factors that influence regional glucose uptake, such as tumor necrosis or site-specific differences in perfusion, might further complicate the use of ^18^F-FDG uptake for predictive purposes. The balance of all these factors makes up the ^18^F-FDG uptake of a tumor lesion.

It is questionable if the observed correlation between SUV, MTV, and TLG and clinical outcome in patients treated with checkpoint inhibitors is due to the immunotherapy treatment. It has been shown that high total tumor ^18^F-FDG uptake expressed as TLG or MTV correlates with poor prognosis in patients with NSCLC, independent of disease stage or treatment modality [39,40,41,42,43]. Although a high SUVmax seems to correlate with high PD-L1 expression on tumor cells, a large metabolic tumor volume can be a stronger negative prognostic parameter. Therefore, ^18^F-FDG-PET-CT derived parameters exhibit both prognostic and predictive value, and ^18^F-FDG uptake is being influenced by immunostimulatory and immunosuppressive cells as well as tumor cells. PET tracers with more specific target binding to cells or receptors involved in tumor immunology are potentially more useful in predicting treatment response to immunotherapy.

### 2.2. PD-(L)1 PET

With the use of radiolabeled drugs or drug targets, the biodistribution of the drug and target expression in tumor lesions can be visualized and quantified with PET, as demonstrated earlier for targets other than PD-1 and PD-L1. For example, in patients with metastatic breast cancer, Chae et al. [44] demonstrated that visualization and quantification of estrogen receptor expression with 16α-[¹⁸F]fluoro-17β-oestradiol (^18^F-FES) PET-CT is feasible. A strong correlation was observed between ¹⁸F-FES PET-CT and estrogen receptor expression by IHC. A negative estrogen receptor status agreement of 100% was found between ^18^F-FES PET-CT and IHC. Another study visualized the mechanism of action of everolimus, a mechanistic target of rapamycin (mTOR) inhibitor that reduces the amount of vascular endothelial growth factor (VEGF). They used bevacizumab labeled with Zirconium-89 (^89^Zr), a VEGF-A-binding antibody tracer [45]. In patients with clear renal cell carcinoma, a significant decrease in ^89^Zr-bevacizumab tumor uptake was observed after treatment initiation with everolimus. These examples clearly demonstrate the concept and feasibility of imaging drug targets using PET, and this concept is currently under investigation in the field of checkpoint inhibitor therapy. Preclinical work has shown that ^18^Fluor-labeled anti-PD-L1 adnectin (^18^F-BMS-986192) and ^89^Zirconium-labeled nivolumab (^89^Zr-nivolumab) were able to visualize PD-L1 and PD-1 expression of tumors [46,47]. In humans, these tracers were evaluated in a prospective study in NSCLC patients treated with nivolumab (Table 1) [48]. With both tracers, adequate tumor-to-background contrast was found, and ^18^F-BMS-986192 uptake demonstrated substantial heterogeneity between patients, between tumor lesions of the same patient, and even within tumor lesions. Furthermore, ^18^F-BMS-986192 uptake correlated with PD-L1 expression by IHC, and patients with a treatment response showed a higher ^18^F-BMS-986192 uptake compared to patients without a response. A larger trial is currently recruiting patients in our institution to evaluate the predictive power of this tracer (NCT03564197, Figure 1) in patients with NSCLC. Furthermore, Xing et al. used ^99m^Tc-NM-01 (^99m^Tc labeled single-domain antibody against PD-L1) for SPECT imaging in NSCLC patients and found a correlation between tumor tracer uptake and PD-L1 IHC results [49]. Bensch et al. performed imaging in 22 patients with NSCLC, metastatic bladder cancer, or triple-negative breast cancer [19]. These patients were imaged with ^89^Zr-atezolizumab prior to atezolizumab treatment. Their results are comparable with the study from Niemeijer et al. [48]. All known metastatic sites were visualized, and heterogeneity was observed in the 20 patients with more than one lesion. Heterogeneous intra-tumor tracer uptake was observed in multiple lesions as well. The quantity of ^89^Zr-atezolizumab uptake correlated with the outcome of atezolizumab treatment. Regardless of tumor pathology or total tumor burden, tumor tracer uptake correlated with the best tumor response category according to RECIST 1.1. Importantly, ^89^Zr-atezolizumab uptake correlated better with clinical response than PD-L1 IHC.

### 2.3. CD8-PET

Another approach in response prediction with PET is imaging CD8+ T cells. It is well known that T-cell inflamed tumors are associated with a favorable response to immunotherapy compared to the non-T-cell-inflamed phenotype [51]. An early influx of CD8+ cells could be useful as a readout of the intended treatment effect. Seo et al. and Tavare et al. [52,53] showed the capability for PET to visualize and quantify CD8+ T cells in tumor and non-tumor tissues in multiple mouse models. The results of the first in-human study have been published recently [50]. Six patients with solid tumors were scanned with ^89^Zr-IAB22M2C, a CD8-specific tracer, and the observed biodistribution of this tracer suggested successful imaging of CD8+ T cells. Data on the predictive value of this approach are expected in the near future.

## 3. Early Response Evaluation with PET during Treatment with Checkpoint Inhibitors

Traditionally, treatment response in solid tumors is evaluated using a morphological approach such as RECIST 1.1, based on CT imaging [20]. To correct for the different potential patterns of response in checkpoint inhibitor therapy, the irRC, and iRECIST criteria were developed [26,54]. However, in 1999, the EORTC response criteria for ^18^F-FDG-PET were already published, and these recognized that a metabolic response is seen early after treatment initiation was likely to be important [55]. In 2009 Wahl et al. proposed the PET response criteria in solid tumors (PERCIST) (Table 2) [56].

In 2015, the EORTC criteria were applied to a cohort of melanoma patients treated with ipilimumab, a CTLA-4 blocking monoclonal antibody [59]. The investigators concluded that response evaluation with ^18^F-FDG-PET after only two cycles of ipilimumab was predictive of the final treatment response in patients with progressive metabolic disease (PMD) and stable metabolic disease (SMD). Patients with a best-observed response of partial metabolic response (PMR) showed pseudoprogression on the early evaluation scan and were therefore misclassified. Similarly, the PERCIST criteria were evaluated in small studies to investigate its predictive power in the checkpoint inhibitor therapy setting [60,61,62]. These studies demonstrated that ^18^F-FDG-PET has the potential to identify treatment response early after treatment initiation, but analogous to the EORTC criteria, the PERCIST criteria also did not discriminate between pseudoprogression and actual disease progression. To correct these new response patterns, the immunotherapy modified PET response criteria in solid tumors (imPERCIST) were introduced [63]. While in PERCIST, the appearance of a new lesion always indicates PMD, in imPERCIST, this is only the case if the intensity of ^18^F-FDG uptake for measured lesions increases by at least 30%. In the study from Ito et al., the imPERCIST criteria showed a higher correlation between ^18^F-FDG-PET results and survival compared to the PERCIST criteria in a cohort of melanoma patients treated with ipilimumab [63].

To combine the strengths of ^18^F-FDG-PET and CT in response evaluation in checkpoint inhibitor therapy, Cho et al. performed a study in a small cohort of 20 melanoma patients [57]. In this study, RECIST 1.1, irRC, the EORTC criteria, and PERCIST were all applied and used to propose new criteria: PET/CT Criteria for early prediction of Response to Immune checkpoint inhibitor therapy (PECRIT) to predict clinical response to checkpoint inhibitor therapy more accurately. The authors state that with an ^18^F-FDG-PET/CT scan performed 3 to 4 weeks after treatment initiation, the PECRIT derived sensitivity, specificity, and accuracy to predict response at 4 months were 100%, 93.3%, and 95.0%, respectively. However, there are no independent prospective data available yet to validate these criteria.

Another study that investigated the role of ^18^F-FDG-PET for response evaluation was performed by Anwar et al. [58]. A cohort of 41 metastatic melanoma patients treated with ipilimumab was evaluated with ^18^F-FDG-PET after two treatment cycles. The results of the early ^18^F-FDG-PET were related to the clinical response after the end of the whole ipilimumab treatment. The investigators found no difference in SUVmax or SUVmean of target lesions between the response groups but did conclude that the number of new ^18^F-FDG-avide lesions on PET was a very suitable predictor of the patient’s clinical response. A cutoff of four new lesions was found to provide a suitable indication of treatment failure in this patient cohort. Based on these results, the PET Response Evaluation Criteria for Immunotherapy (PERCIMT) were proposed. In this criteria, four or more lesions of less than 1.0 cm, three or more lesions of more than 1.0 cm, or two or more new lesions of more than 1.5 cm define a progressive disease.

Castello et al. applied most of the above-mentioned response evaluation criteria in a cohort of NSCLC patients [64]. The RECIST 1.1, imRECIST, EORTC, PERCIST, imPERCIST, and PERCIMT criteria were all applied in a cohort of 52 NSCLC patients treated with nivolumab or pembrolizumab. The authors concluded that PET-based criteria were more reliable than diameter measurements in the detection of treatment response and that the imPERCIST criteria had the best performance in predicting treatment response and survival. It is important to mention, however, that the assessment of all the criteria in this study was made 8 weeks after treatment initiation, making it difficult to compare these results to those obtained in studies where response evaluation was performed 3–4 weeks after the start of treatment.

These data indicate that ^18^F-FDG-PET has the potential to have additional value over CT-based treatment evaluation alone. With all these different response criteria (Table 1), it is difficult to determine which criteria has the strongest predictive value during checkpoint inhibitor therapy. All of these criteria are based on small cohorts with either NSCLC or melanoma patients. Some results are obtained during anti-CTLA-4 therapy, while others during anti-PD-1 treatment. Most importantly, none of these criteria were prospectively validated on a large patient group. There is no doubt, however, about the unmet need for a reliable response evaluation method early in the course of treatment initiation with checkpoint inhibitors. Early recognition of an ineffective treatment regime enables patients to derive the benefits from another line of anticancer therapy, limits potential toxicity, and reduces the economic impact of immunotherapy.

## 4. Value of ^18^F-FDG-PET in Late Response /Residual Disease Evaluation

The arguments of potential toxicity and economic impact also apply to the evaluation of residual disease after an earlier identified treatment response. Does the residual lesion on the CT contain viable tumor cells, scar tissue, or lymphoid tissue? Is additional therapy indicated? Figure 2 shows a clear example of the additional information provided by ^18^F-FDG-PET. The optimal treatment duration of checkpoint inhibitors is actively being studied, and non-invasive biomarkers can potentially identify patients that can benefit from shorter courses of immunotherapy [65,66,67,68,69].

In an attempt to predict the long-term outcome, Tan et al. performed a retrospective analysis of metastatic melanoma patients treated with nivolumab [70]. An ^18^F-FDG-PET and CT scan made one year after treatment initiation was analyzed using modified EORTC criteria* and RECIST 1.1, respectively. All of the patients with CR on CT had CMR on the ^18^F-FDG-PET, and none of these patients developed disease recurrence during follow-up. Of the 69 patients with PR on CT, 47 had a CMR. In the SD group, two out of six had a CMR. Follow-up data showed that CMR predicts a favorable outcome; 100% of patients with CMR were progression-free at one year post imaging compared to 57% of the patients without CMR. 

* Modifications to EORTC criteria:SUV max of the five most intense metastatic lesions was measured at baseline and on the 1-year PET;CMR is similar or lower radiotracer uptake than the mediastinal blood pool;Bilateral lymphadenopathy with radiotracer uptake on PET was considered a benign sarcoid-like pattern.

A study by Kong et al. showed comparable results with respect to the correlation between residual metabolic activity and clinical outcome [71]. Additionally, they performed biopsies in patients with remaining or new ^18^F-FDG-avid lesions. Eight ^18^F-FDG-avid lesions were biopsied or resected: four in patients with PD, three with SD/PR-group, and one with CR (all according to RECIST 1.1). In three biopsies, inflammatory infiltrates were found instead of tumor cells. A trend was observed, namely that a higher SUVmax correlated with residual disease activity, unlike patients with false-positive uptake (median SUVmax 18 vs. 7.1).

These results indicate that even after more than a year following the first administration of immunotherapy, response evaluation can be challenging. In addition to conventional CT, ^18^F-FDG-PET can provide valuable information in patients with residual or new lesions to be able to differentiate between fibrotic tissue and viable tumor cells (Figure 3). When a CMR is seen, the continuation of treatment might not be indicated, while in the case of persistent metabolic activity, a biopsy can differentiate between viable tumor cells or immune infiltrate and guide treatment decisions. This approach of determining the treatment duration should, however, be explored in prospective trials. Considering the difference in efficacy of immunotherapy in melanoma patients compared to other solid malignancies, extrapolation of these results to other tumor types should be performed with caution.

## 5. Radiomics

The concept of using quantitative imaging data in response prediction can be refined with the use of radiomics. Large amounts of data are obtained with routinely performed imaging procedures, and hundreds of imaging features (shape, texture, etc.) can be integrated into statistical models, whether or not combined with clinical data. Applying machine learning and artificial intelligence to this data is an active field of research, and early reports have shown that radiomics have prognostic value and may capture gene expression in lung and head-and-neck cancer [72,73]. In the field of immunotherapy, this concept can be applied to response prediction with PET. There are only a limited number of studies investigating radiomics in PET in patients treated with immunotherapy. Mu et al. combined radiomics features of baseline CT, PET, and PET-CT fusion images to predict durable clinical benefit in NSCLC patients treated with immunotherapy [74]. They applied a multiparametric radiomics signature on PET-CT data obtained before treatment initiation. Compared to clinical features such as ECOG status, histology, and distant metastasis, their radiomics signature had additive value to predict durable clinical benefit. Furthermore, the authors conclude that the strength of their model was a result of the integration of radiomics features of both CT and PET. One important shortcoming, however, was the unavailability of PD-L1 status in the majority of the included patients. A comparison of their model with the currently available biomarker, PD-L1 expression, was therefore not possible. Valentinuzzi et al., however, were able to compare a radiomics model to PD-L1 expression level [75]. They concluded that their model outperformed the PD-L1 tumor proportion score in predicting response (AUC 0.90 compared to 0.60) in a patient population with a PD-L1 TPS > 50%, eligible for pembrolizumab monotherapy. Furthermore, Polverari et al. performed a retrospective study in 57 NSCLC patients that were treated with immunotherapy in a first and second-line treatment setting [76]. The investigators demonstrated that patients with radiomics features such as high tumor volume, TLG, and heterogeneity expressed by skewness and kurtosis (measurements that express how heavily the tails of distribution differ from the tails of a normal distribution) had a higher probability of treatment failure.

These results indicate that radiomics can have added value to other biomarkers, but as with any other biomarker, radiomics have their limitations. For example, large variability in acquisition and reconstruction methods exists between different scanners and institutions, complicating the applicability of this technique in clinical practice. Furthermore, since automatic segmentation of tumor lesions is not perfect, the segmentation of tumors can be very labor-intensive and time-consuming. As these limitations can be overcome with technical improvements in this relatively new approach, it is important that larger prospective trials will be carried out to evaluate the true added value of radiomics in guiding treatment decisions.

## 6. Discussion

The need for better predictive biomarkers is high and will increase in the near future as more and more immunotherapy treatments, treatment indications, and treatment combinations will emerge. A one size fits all treatment approach is suboptimal. To this point, as is also stated in an extensive review by Garcia-Figueiras et al., PET has a high potential for clinicians to make well-founded decisions before and during immunotherapy treatment [77]. In the current review, we described how the available data can be interpreted and how it can help clinicians to make treatment decisions.

The results for response prediction with ^18^F-FDG-PET indicate that the predictive value of a single baseline scan is low and inconsistent. Higher metabolism of tumor and healthy tissue is non-specific and can be related to opposite predictive factors such as PD-L1 expression, CD8 infiltration, infiltration of regulatory T cells, and high tumor proliferation rate. The use of artificial intelligence (radiomics) in ^18^F-FDG-PET might be able to overcome some of these problems, but this remains uncertain at this moment [78].

As ^18^F-FDG fails to predict response, PET scans with specific immune tracers show promising results (Table 2). The first clinical studies have proven the feasibility of imaging and quantifying PD-(L)1 expression. Earlier mentioned studies with ^89^Zr-atezolizumab, ^89^Zr-nivolumab, and ^18^F-BMS-986192 showed that in a small group of solid cancer patients (mainly NSCLC), favorable outcome to treatment with checkpoint inhibition when high uptake of the tracer in the tumor was observed. However, the true theragnostic potential of these tracers has to be determined in larger trials, which are currently ongoing (NCT04006522, NCT03514719, NCT03829007, and NCT03564197). Awaiting these results, one important aspect that has to be kept in mind is that the success of immunotherapy does not depend solely on the binding of the checkpoint inhibitor to the PD-(L)1 receptor but also on the tumor microenvironment as a whole and multiple host factors [79].

One important aspect of immune-PET is the choice of a suitable radionuclide. Anti-PD-(L)1 inhibitors are large molecules that are characterized by slow tissue penetration. To be able to obtain adequate images, these monoclonal antibodies (mAbs) should be labeled with a radionuclide with a long half-life. ^89^Zirconium, for example, has a half-life of 78.4 h, compared to 109.8 min for ^18^F that is used in conventional ^18^F-FDG-PET. From a clinical and logistical point of view, this is an important difference. Due to slow tissue penetration, the scans with ^89^Zr were obtained on day 3 and 6, or day 4 and 7 after tracer injection [19,48]. Small anti-PD-L1 protein tracers are under research [80]. These tracers penetrate and bind, due to their size, much quicker in the target tissue. This allows scanning several minutes or hours after tracer injection, which enables labeling with ^18^F. This is more patient-friendly and easier to interpret than ^89^Zr, in particular, because of the relatively lower noise level and higher resolution of ^18^F-PETscans [81].

Not only prediction before administration of immunotherapy has room for improvement. Early response evaluation also leaves much to be desired. The majority of immunotherapy trials still base their response evaluation on CT scans only [1,2,3,4,5,6,7,8,9,10,11,12]. When using CT alone with iRECIST criteria, it is common to proceed with treatment with immunotherapy when progression is seen, as this might be the result of pseudoprogression. However, as previously mentioned, this phenomenon is rare, and many patients will continue their treatment in the presence of disease progression. Of all the immunotherapy adjusted response criteria in PET, the imPERCIST criteria seem to be the strongest at this point. It shows predictive strength in a small independent cohort and does not show conflicting results in other studies. There is a need for a large prospective study in which these (and preferably all) criteria are being evaluated and validated.

Building on the successes of immunotherapy trials that led to multiple FDA and EMA approvals, numerous trials are now exploring combinations of various immunotherapies. Not only the combination of drugs but also the timing and sequence might be crucial [82]. Using PET-derived measures of the tumor microenvironment prior to and shortly after treatment initiation with immunotherapy, a non-invasive, early readout of intended biological effect can be obtained, for example, induced PD-L1 expression or CD8+ T-cell influx. This approach can be helpful for drug development purposes. In the near future, clinicians should be able to use a combination of data, such as immune-PET, ^18^F-FDG-PET, PD-L1 IHC, TMB, tumor-infiltrating lymphocytes, and peripheral blood mononuclear cells. Based on this data, a patient-tailored treatment will be designed for each individual patient.

## 7. Conclusions

Predicting and evaluating a treatment response during immunotherapy is challenging. RECIST might underestimate response to checkpoint inhibition and adjusted criteria have been developed that are superior, but still are suboptimal. ^18^F-FDG-PET has clear added value to CT alone in response evaluation, especially in the case of residual disease. However, the PET based evaluation criteria need validation in large cohorts before they can be applied in clinical practice. Checkpoint/immune cell PET has been validated for clinical use and correlate with response in small number studies. Future studies have to show whether they can actually guide clinicians or assist in future drug development. 

## Figures and Tables

**Figure 1 cancers-13-03083-f001:**
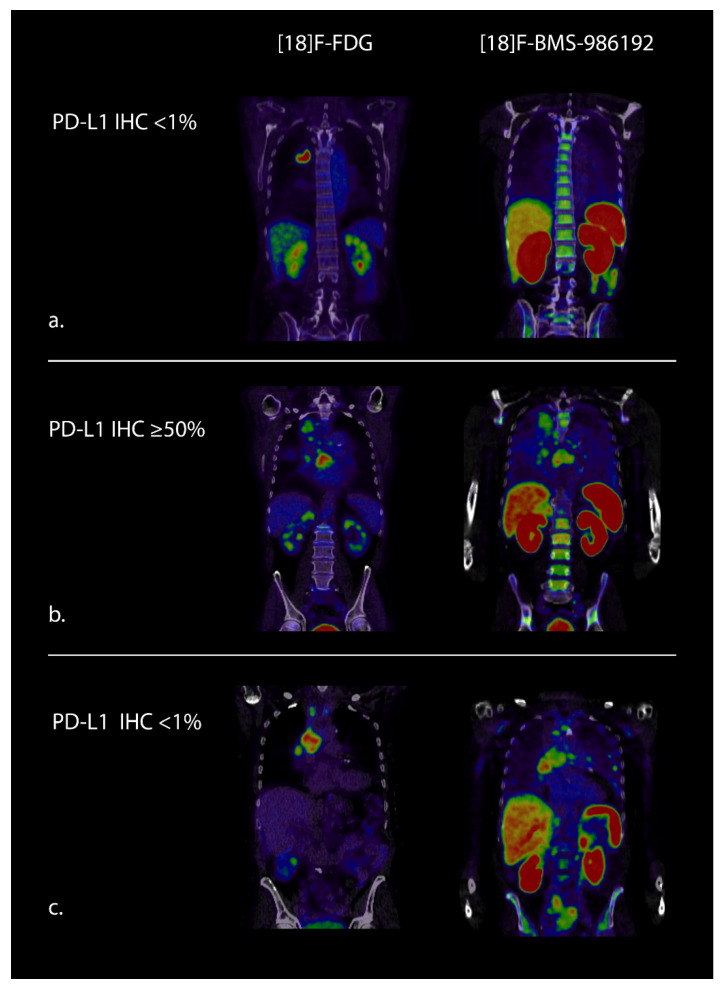
^18^F-FDG-PET and ^18^F-BMS-986192 PET (PD-L1 directed tracer) images of (**a**) a patient without PD-L1 expression and no tracer uptake in the tumor on the PD-L1 PET. (**b**) A patient with high PD-L1 expression and high tracer uptake in the tumor on the PD-L1 PET and (**c**) a patient without PD-L1 expression according to IHC, but tracer uptake in tumor lesions on the PD-L1 PET. These examples demonstrate that imaging results correspond with IHC (**a**,**b**) but can also reveal possible sampling error or tumor heterogeneity (**c**). These images were obtained in our institution as part of the NCT03564197 trial.

**Figure 2 cancers-13-03083-f002:**
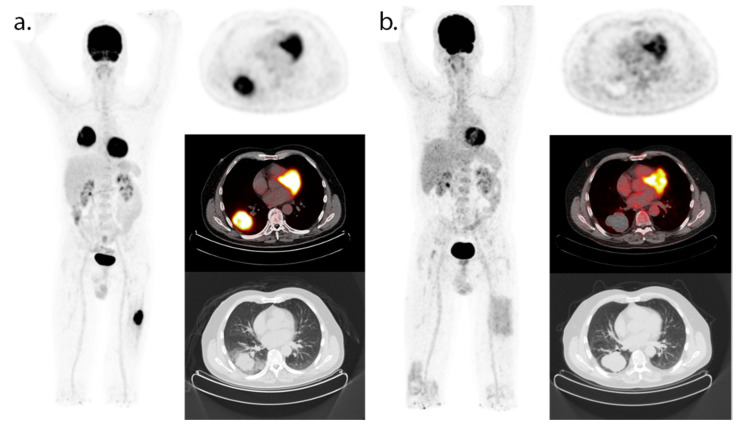
^18^F-FDG-PET/CT (maximum intensity projection and axial PET, fused PET/CT and CT images) at baseline (**a**) and during follow-up of immunotherapy (**b**). These images were obtained in our clinical institution as part of routine clinical follow-up. The patient provided verbal consent for the use of this anonymized imaging data. Baseline FDG-PET/CT prior to the start of nivolumab treatment showed a primary NSCLC tumor located in the right lower lobe and metastasis in the left femur. FDG-PET/CT after 13 months of nivolumab treatment showed a complete metabolic response. Due to toxicity, the patient has stopped nivolumab treatment in July 2017 and is still in remission.

**Figure 3 cancers-13-03083-f003:**
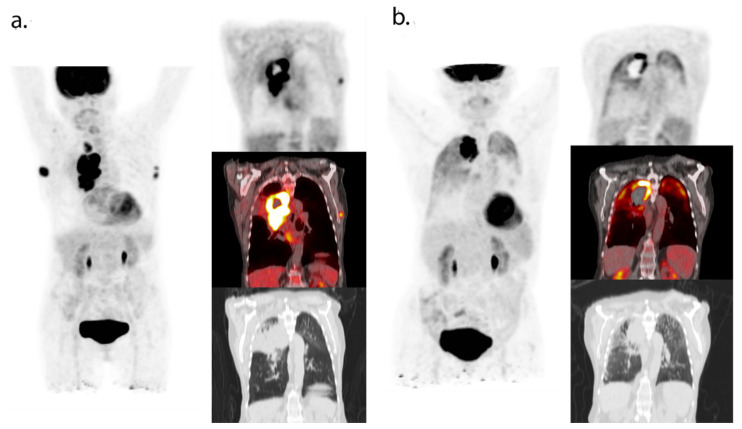
^18^F-FDG-PET/CT (maximum intensity projection and coronal PET, fused PET/CT and CT images) at baseline (**a**) and during follow-up of immunotherapy (**b**). These images were obtained in our clinical institution as part of routine clinical follow-up. The patient provided verbal consent for the use of these anonymized imaging data. At baseline, ^18^F-FDG-PET/CT showed an NSCLC tumor in the right lung and lymph node metastases in the left and right axilla. The patient was treated with pembrolizumab monotherapy and obtained a partial response. ^18^F-FDG-PET/CT six months after initiation of therapy showed oligoprogression in the right upper lobe. No new disease activity was observed in the lymph nodes. The patient received radiotherapy on the tumor located in the right lung and remained in remission.

**Table 1 cancers-13-03083-t001:** Main results of clinical PET studies using immune tracers.

Tracer (Target)	Injected Dose	Timing Image Acquisition ^1^	Study Population	Main Results
^18^F-BMS-986192 (PD-L1) [48]				Tracer uptake correlates with PD-L1 expression by IHC
	3 MBq/kg ±10%	1 h post-injection	NSCLC 13 patients	
				Lesional tracer uptake is related to response
89Zr-nivolumab (PD-1) [48]	37 MBq ± 10%	7 days post-injection	NSCLC 13 patients	Tracer uptake correlates with aggregates of PD-1 determined by IHCLesional tracer uptake is related to response
^99m^Tc-NM-01 (PD-L1) [49]	Group 1 (3.8–8.4 MBq/kg) Group 2 (9.1–10.4 MBq/Kg)	2 h post-injection	NSCLC 16 patients	Tracer uptake correlates with PD-L1 expression by IHC
^89^Zr-atezolizumab (PD-L1) [19]	37 MBq	7 days post-injection	NSCLC 9 patients Bladder cancer 9 patients Breast cancer 4 patients	Tracer showed a stronger correlation with clinical response compared to PD-L1 IHC
^89^Zr-IAB22M2C (CD8+ T-cell) [50]	Mean 108 (range 92–120) MBq	1–2 days post-injection	NSCLC 4 patients Melanoma 1 patientHepatocellular cancer 1 patient	Biodistribution suggests successful targeting of CD8+ T cells

^1^ The differences between the tracers with respect to the timing of imaging acquisition are due to the size of the tracer used. Large antibodies, such as nivolumab and atezolizumab, have a slow tissue penetration and plasma clearance compared to smaller tracers, such as BMS-986192 and IAB22M2C. Therefore, adequate images can be obtained earlier using smaller tracers compared to the larger antibodies.

**Table 2 cancers-13-03083-t002:** Overview of response evaluation criteria.

Imaging Characteristics	CT		^18^F-FDG-PET(-CT)				
	RECIST 1.1 [20]	iRECIST [21,26]	EORTC [55]	PERCIST [56]	PERCRIT [57]	PERCIMT [58]	imPERCIST [36]
Target lesion at baseline	Maximum two lesions per organ and five lesions total	Per RECIST 1.1	All FDG-avid lesions	Hottest lesion(s). Maximum two per organ	Per RECIST 1.1	Per PERCIST	Per PERCIST
Non-target lesion	Contribute to the CR, PR, SD, and PD	Contribute to the iCR, iPR, iSD, and iPD	-	Contribute to the CMR, PMR, SMD, and PMD	Contribute to the CR, PR, SD and PD	Contribute to the CMR, PMR, SMD, and PMD	Contribute to the CMR, PMR, SMD, and PMD
New lesion	Always represent PD	iUPD, require a next imaging assessment to confirm	Always represent PMD	Always represent PMD	Always represent PD	Number and size of lesions define CR/PR/SD or PD	Does not represent PMD
CR/CMR	Disappearance of all target and non-target lesions	iCR: per RECIST 1.1 at first or at the next assessment within 4–8 weeks after iUPD	Complete resolution of FDG uptake within tumor volume	Complete resolution of FDG uptake and disappearance of all other lesions	Per RECIST 1.1	Per PERCIST	Per PERCIST
PR/PMR	≥30% decrease in the sum of diameters of target lesions. Persistence of one or more non-target lesion(s)	iPR: per RECIST 1.1 at first or at the next assessment within 4–8 weeks after iUPD	Reduction of 15–25% in tumor SUV after 1 cycle of therapy an d > 25% after more than 1 cycle of therapy	>30% relative decrease and >0.8 absolute decrease in SULpeak of hottest lesion	Per RECIST 1.1	If the sum of SULpeak decreased by at least 30% and >0.8 absolute decrease in SULpeak of hottest lesion.	If the sum of SULpeak decreased by at least 30% and >0.8 absolute decrease in SULpeak of hottest lesion
SD/SMD	Neither sufficient shrinkage to qualify for PR nor sufficient increase to qualify for PD	iSD: per RECIST 1.1 at first or at the next assessment within 4–8 weeks after iUPD	Not meeting criteria for CMR, PMR, or PMD	Not meeting criteria for CMR, PMR, or PMD	Not meeting criteria for CR/PR or PD	Not meeting criteria for CMR, PMR, or PMD.	Not meeting criteria for CMR, PMR, or PMD
PD/PMD	≥20% increase in sum of diameters of target lesion(s) or unequivocal progression of non-target lesion(s) or appearance of new lesion(s)	iUPD: per RECIST 1.1. iCPD: iUPD and confirmed 4–8 weeks later	>25% increase within tumor region, visible increase in extent of FDG uptake (20% in longest dimension), or appearance of new FDG positive lesions	>30% relative increase and >0.8 absolute increase in SULpeak of hottest lesion (s) or unequivocal progression of FDG-avid non-target lesion or appearance of new FDG-avid lesion(s)	Per RECIST 1.1	≥4 new lesions of less than 1.0 cm in functional diameter; or ≥3 new lesions of more than 1.0 cm in functional diameter; or ≥2 new lesions of more than 1.5 cm in functional diameter.	>30% increase and >0.8 absolute increase in SULpeak, from baseline scan in a pattern typical of tumor and not of infection/treatment effect

(i)CR = (immune) complete response; (i)PR = (immune) partial response; (i)SD = (immune) stable disease; (i)PD = (immune) progressive disease; CMR = complete metabolic response; PMR = partial metabolic response; SMD = stable metabolic disease; PMD = Progressive metabolic disease; iUPD = immune unconfirmed progressive disease; iCPD = immune confirmed progressive disease; TL = target lesion(s).

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
