# Peer review of "Response Prediction and Evaluation Using PET in Patients with Solid Tumors Treated with Immunotherapy"

_cancers, 2021, doi:10.3390/cancers13123083_

Round 1

Reviewer 1 Report

This is a review regarding response prediction to immunotherapy using PET. There are several main issues:

A material and methods section is missing. It is not clear therefore the nature of this review and if any criteria was used to select paper.  

A very brief resuming table regarding the main improvements of new criteria (imPERCIST, imRECIST, and so on) may be useful, magnifyng paper’s readability.

Also tables resuming the main studies cited and the main characteristics of tracers and molecules cited are warranted.

Please, specify in the text if figures were taken from the literature or from Authors’ own Institution

As regards the radiomics part, why do the authors selected only those 2 studies?

Discussion section is definetively somehow lacking and not well structured. Please rephrase also considering potential theragnostic applications of such molecules.

The whole paper need at least a minor English Scientific Language revision

“Grizzi et al reported the preliminary analysis of 27 patients with NSCLC treated with checkpoint inhibotors[32].”  Please correct with “inhibitors”.

“An approach to assess and size and uptake of 18F-FDG at the same time in malignant lesions”. This may be misleading, please rephrase.

“The PET outcome measurements also correlated with outcome to atezolizumab treatment – tumor tracer uptake correlated with the best tumor response category according to RECIST 1.1, irrespective of tumor type and tumor load.” Not clear, please rephrase.

MAbs, TILs, and PBMCs acronyms are missing

“Larger prospective studies ongoing (NCT04006522, NCT03514719, NCT03829007 and NCT03564197) to further investigate the role of immune-PET in re- sponse prediction.” At least, a verb is missing. Please, pay attention.

Please, uniform text character in the whole manuscript and abstract.

Author Response

Comments and Suggestions for Authors

This is a review regarding response prediction to immunotherapy using PET. There are several main issues:

Dear reviewer,

We thank you for your in-depth review. Below you can find your comments and suggestions with our corrections to these remarks explained in red underneath. 

A material and methods section is missing. It is not clear therefore the nature of this review and if any criteria was used to select paper.  

We thank the Reviewer for this suggestion. Accordingly we have now added a paragraph of Evidence acquisition to replace the classic Materials and methods paragraph of this  non-systematic review.

A very brief resuming table regarding the main improvements of new criteria (imPERCIST, imRECIST, and so on) may be useful, magnifyng paper’s readability.

We thank Reviewer for this great suggestion. We have created a table to summarize and compare the specific features of the evaluation criteria. Please see Table 2, entitled ‘Overview of response evaluation criteria’

Also tables resuming the main studies cited and the main characteristics of tracers and molecules cited are warranted.

According to the Reviewer’s suggestion we have created a new table with the tracer characteristics and main results (Table 1). In this table, and the table with the evaluation criteria, we’ve added references to highlight the main cited studies.

Please, specify in the text if figures were taken from the literature or from Authors’ own Institution

The requested information has been provided. All figures are taken from our own Institution. This information is now added to the main text or figure legend.

As regards the radiomics part, why do the authors selected only those 2 studies?

This part only discussed radiomics specifically in PET and immune checkpoint inhibitor therapy, this is now stated more clearly in the introduction. As a consequence of the reviewer's comment we reviewed the present literature and added another important paper by Polverari et al. to the review.

Discussion section is definetively somehow lacking and not well structured. Please rephrase also considering potential theragnostic applications of such molecules.

We thank the reviewer for the suggestion. We made changes to the discussion to make the structure more clear and added a phrase concerning theragnostic potential of immunotracers.  

The whole paper need at least a minor English Scientific Language revision

The English Scientific Language revision has been applied.

“Grizzi et al reported the preliminary analysis of 27 patients with NSCLC treated with checkpoint inhibotors[32].”  Please correct with “inhibitors”.

We thank you for your thorough review and we apologize for the typo. This has now been corrected.

“An approach to assess and size and uptake of 18F-FDG at the same time in malignant lesions”. This may be misleading, please rephrase.

Thank you, good suggestion. The sentence has been rephrased to make it more clear.  

“The PET outcome measurements also correlated with outcome to atezolizumab treatment – tumor tracer uptake correlated with the best tumor response category according to RECIST 1.1, irrespective of tumor type and tumor load.” Not clear, please rephrase.

Thank you for noticing this, we rephrased the sentence to clarify this statement.

MAbs, TILs, and PBMCs acronyms are missing

The abbreviations are now written out, to make it understandable

“Larger prospective studies ongoing (NCT04006522, NCT03514719, NCT03829007 and NCT03564197) to further investigate the role of immune-PET in re- sponse prediction.” At least, a verb is missing. Please, pay attention.

Thank you for noticing, the sentence has been corrected

Please, uniform text character in the whole manuscript and abstract.

We apologize for the inconsistency, it has been corrected.

Submission Date

22 April 2021

Date of this review

11 May 2021 21:26:31

Reviewer 2 Report

Dear authors,

The topic of your review is very interesting but I think that your manuscript should be improved. 

General comments: 

The PRISMA flow diagram is missing. Has this methodology been used to ensure the exhaustiveness of this review?

What is the origin of the images presented in the figures? If they are from original clinical trials, this should be clearly specified. The authors should complete an ethics statement. If they are from other published studies, a clear reference should be given in the legend.

I understand that the review focuses more on tep imaging, but (page 4) the section dealing with "PD(L)1 PET" starts with estradiol and VEGF receptor imaging. why?

What is the added value of this review compared to this one (RadioGraphics 2020; 40:1987–2010), for instance?

I would suggest instead that the authors present the full results of their clinical trial (NCT03564197, I guess) where the evaluation of the predictive power of the BMS986192 tracer uptake is a very relevant and interesting question.

Minor remarks:

Please, verify this sentence: "Allowing scanning several minutes or hours after tracer injection which enables labeling with 18Fl" (p. 10)

Please, could you give the quantitative data to support your assertion: "The MAJORITY of immuno-therapy trials still base their response evaluation on CT scans only."

Author Response

Dear authors,

The topic of your review is very interesting but I think that your manuscript should be improved. 

Dear reviewer,

We thank you for your in-depth review. Below you can find your comments and suggestions with our corrections to these remarks explained in red underneath. 

General comments: 

The PRISMA flow diagram is missing. Has this methodology been used to ensure the exhaustiveness of this review?

We thank the Reviewer for this suggestion. Since our review is not a systematic review, the PRISMA methodology was not used. However, we added a paragraph, ‘Evidence acquisition’ to address this.

What is the origin of the images presented in the figures? If they are from original clinical trials, this should be clearly specified. The authors should complete an ethics statement. If they are from other published studies, a clear reference should be given in the legend.

The origin of the images from figure 1 is the currently running trial in our institution (ClinicalTrials.gov Identifier NCT03564197) as is stated in the text. We also added this information to the legend to make this more clear. The patient signed informed consent to use his/her data for data analysis and publication (anonymized).  

The images from figure 2 and 3 are obtained in our institution outside a clinical trial. Both patients gave verbal consent to use these images. This information is added to the legend.

I understand that the review focuses more on tep imaging, but (page 4) the section dealing with "PD(L)1 PET" starts with estradiol and VEGF receptor imaging. why?

We thank you for this fair question. These examples were used to introduce the concept and demonstrate the feasibility of imaging drug targets using PET. We’ve clarified this more in the manuscript

What is the added value of this review compared to this one (RadioGraphics 2020; 40:1987–2010), for instance?

The Reviewer addresses an important aspect. Both reviews show similarities, however the focus of our review is slightly different. Our review focusses mainly on the use of PET in response prediction and evaluation and less on other imaging techniques as for example diffusion-weighted MRI and perfusion CT. Furthermore, we do not discuss the mechanism of action of immunotherapy as extensive, resulting in a more compact overview of the use of PET in immunotherapy.

I would suggest instead that the authors present the full results of their clinical trial (NCT03564197, I guess) where the evaluation of the predictive power of the BMS986192 tracer uptake is a very relevant and interesting question.

We thank the reviewer for the shown interest in the results of our clinical trial. The study is still ongoing, so at present, the results are not yet available. Once the study will be finished, the publication of the results will follow.

Minor remarks:

Please, verify this sentence: "Allowing scanning several minutes or hours after tracer injection which enables labeling with 18Fl" (p. 10)

This sentence is verified and corrected.

Please, could you give the quantitative data to support your assertion: "The MAJORITY of immuno-therapy trials still base their response evaluation on CT scans only

Quantitative data to support this assertion is not available. We have now added references of important clinical trials that used RECIST to support this assertion.

Round 2

Reviewer 1 Report

Paper is ready for publication

Author Response

We would like to thank the reviewer for the comments and suggestions. 

Reviewer 2 Report

Dear authors,

Thank you for responding to the comments.
However, I believe that presenting the results of your clinical study as an original article would have been more appropriate. As you note there are many critical reviews similar to yours in the field.

Author Response

We would like to thank the reviewer for the comments and suggestions. As soon as the results of our clinical trial are available, we will present this in a seperate manuscript.